# PUSHING THE BOUNDS OF DROPOUT

## ABSTRACT

We show that dropout training is best understood as performing MAP estimation concurrently for a family of conditional models whose objectives are themselves lower bounded by the original dropout objective. This discovery allows us to pick any model from this family after training, which leads to a substantial improvement on regularisation-heavy language modelling. The family includes models that compute a power mean over the sampled dropout masks, and their less stochastic subvariants with tighter and higher lower bounds than the fully stochastic dropout objective. We argue that since the deterministic subvariant's bound is equal to its objective, and the highest amongst these models, the predominant view of it as a good approximation to MC averaging is misleading. Rather, deterministic dropout is the best available approximation to the true objective.

## 1 INTRODUCTION

The regularisation technique known as dropout underpins numerous state-of-the-art results in deep learning (Hinton et al. 2012; Srivastava et al. 2014), and its application has received much attention in the form of optimisation (Wang & Manning 2013) and attempts at explaining or improving its approximation properties (Baldi & Sadowski 2013; Zolna et al. 2017; Ma et al. 2016). The dominant perspective today views dropout as either an implicit ensemble method (Warde-Farley et al. 2013) or averaging over an approximate Bayesian posterior (Gal & Ghahramani 2016a). Regardless of which view we take, dropout training is carried out the same way, by minimising the expectation of the loss over randomly sampled dropout masks. However, at test time these views naturally lead to different algorithms: the Bayesian approach computes an arithmetic average as it marginalises out the weight uncertainty, while the ensemble approach typically uses the geometric average due to its close relationship to the loss. Collectively they are called MC dropout and neither is clearly better than the other (Warde-Farley et al. 2013). A third way to make predictions is to "turn dropout off", that is, propagate expected values through the network in a single, deterministic pass. This deterministic (also known as *standard*) dropout in considered to be an excellent approximation to MC dropout.

This situation is unsatisfactory as it does not provide theoretical grounding for dropout, without which the choice of dropout variant remains arbitrary. In this paper, we provide such theoretical foundations. First, we prove the dropout objective to be a common lower bound on the objectives of a family of infinitely many models. This family includes models corresponding to the three aforementioned methods of evaluation: the arithmetic averaging, the geometric averaging, and the deterministic. Thus by maximising the dropout objective we get a single set of parameters and many models that all have the same parameters but differ in how they make predictions. This allows us to train once and perform model selection at validation time by evaluating the different methods of making predictions corresponding to individual models in the family. Second, we turn the conventional perspective on its head by showing that while dropout training performs stochastic regularisation, the trained model is best viewed as deterministic, not as a stochastic model with a deterministic approximation.

This paper is structured as follows. In §2, we revisit variational dropout (Gal & Ghahramani 2016a) and demonstrate that, despite common perception, sharing of masks is not necessary, neither in theory nor in practice. Then, by recasting dropout in a simple conditional form, we highlight the counterintuitive role played by the variational posterior. §3 contains our main contributions. Here we construct a family of conditional models whose MAP objectives are all lower bounded by the usual dropout objective, and identify a member of this family as best in terms of model fit. In §4, we select the best of this family in terms of generalisation to improve language modelling. Finally, creating a cheap approximation to the bias of this model allows us to get better results from model tuning.

## 2 VARIATIONAL DROPOUT

Since its original publication (Hinton et al. 2012), dropout had been considered a stochastic regularisation method, implemented as a tweak to the loss function. That was until Gal & Ghahramani (2016a) grounded dropout in much-needed theory. Their subsequent work (Gal & Ghahramani 2016b) focused on RNNs, showing that if dropout masks are shared between time steps, the objective for their proposed variational model is the same as the commonly used dropout objective with an $\ell_2$ penalty. Their method became known as **variational dropout**, not to be confused with Kingma et al. (2015), and is used in state-of-the-art sequential models (Merity et al. 2017; Melis et al. 2017). Before we move on to a more general formulation we revisit it to better understand its critical features.

First, we recall the derivation of variational dropout. Consider an RNN that takes input $x$ and maps it to output $y$ and is trained on a set of $N$ data points in paired sets $X, Y$. A variational lower bound on the log likelihood is obtained as follows:

$$
\begin{aligned}
\ln p(Y|X) = \ln \mathop{\mathbb{E}}_{\omega \sim q(\omega)} \frac{p(Y|X,\omega)p(\omega)}{q(\omega)} \\
\geqslant \mathop{\mathbb{E}}_{\omega \sim q(\omega)} \ln p(Y|X,\omega) - \mathrm{KL}(q(\omega)||p(\omega)) \\
= \int q(\omega) \ln p(Y|X,\omega)d\omega - \mathrm{KL}(q(\omega)||p(\omega)) \\
= \sum_{i=1}^{N} \int q(\omega) \ln p(y_i|x_i,\omega)d\omega - \mathrm{KL}(q(\omega)||p(\omega)),
\end{aligned}
\tag{1}
$$

where $p(y|x,\omega)$ is defined by the RNN with weights $\omega$. Variational Bayesian methods then maximise this lower bound with respect to the variational distribution $q(\omega)$. For variational dropout, $q(\omega)$ takes the form of a mixture of two gaussians with small variances: one with zero mean that represents the dropped out rows of weights, and another with mean $\Theta$:

$$
q(\omega_r) = p\mathcal{N}(\omega_r|0,\sigma^2\mathrm{I}) + (1-p)\mathcal{N}(\omega_r|\Theta_r,\sigma^2\mathrm{I})
$$

In the above, $r$ is the index of a row of a weight matrix. Dropping whole rows of weights is equivalent to the more familiar view of dropout over units. The prior over the weights is a zero mean gaussian:

$$
p(\omega) = \mathcal{N}(\omega|0,\sigma_p^2\mathrm{I})
$$

The loss is defined based on Eq. 1. The integrals are approximated using a single sample $\hat{\omega} \sim q(\omega)$, and the KL term is approximated with weight decay on $\Theta$:

$$
\mathcal{L} = -\sum_{i=1}^{N} \ln p(y_i|x_i,\hat{\omega}_i) + \mathrm{KL}(q(\omega)||p(\omega))
\tag{2}
$$

The same dropout mask (and consequently the same $\omega$) is employed at every time step. This sharing of masks is considered the defining characteristic of variational dropout, but we note in passing that the theory for the non-shared masks case is very similar and there is little between them in practice with LSTMs (see Appendix A). With this we conclude the recap of variational dropout, and describe our contributions in the rest of the paper.

### 2.1 DROPOUT AS A CONDITIONAL MODEL

In variational inference the idea is to approximate the intractable and complicated posterior with a simple, parameterised distribution $q$. Crucially, this approximation affects our inferences and predictions. If we are serious about it being an approximation to the posterior and want to reduce its distortion of the model $p$, then $q$ can be made more flexible. But making $q$ more flexible in variational dropout can potentially ruin the regularisation effect. So the particular choice of $q$ plays an important, active role: it effectively performs posterior regularisation and acts as an integral part of the model.

Coming from another angle, Osband (2016) makes the point that in variational dropout the posterior over weights does not concentrate with more data, unlike for example in Graves (2011), which is unexpected behaviour from a Bayesian model. This conundrum is caused by encoding dropout with a

fixed rate mixture of fixed variance components in $q$, which also necessitates expensive tuning of the dropout rate. Gal et al. (2017) proposes a way to address these shortcomings.

To avoid getting bogged down in the issues surrounding the suitability of variational inference and ease interpretation, we construct a straightforward conditional model and lower bound its MAP objective in the same form as the variational objective. Suppose we want to do MAP estimation for the model parameters (the means of the distribution of weights, $\Theta$): $\arg\max_\Theta p(\Theta|X, Y)$. Consider a conditional model $p(Y|X, \Theta)$ as a crippled generative model with $p(x_i)$ constant, $x_i$ and $\Theta$ independent. Place a normal prior on the means $\Theta$ and otherwise make the weights $\omega$ conditional on $\Theta$ the same way as they were in the variational posterior $q(\omega)$:

$$p(\Theta) = \mathcal{N}(\Theta|0, \sigma_p^2 I)$$
$$p(\omega_r|\Theta) = p\mathcal{N}(\omega_r|0, \sigma^2 I) + (1 - p)\mathcal{N}(\omega_r|\Theta_r, \sigma^2 I)$$
$$p(y, \omega|x, \Theta) = p(y|x, \omega)p(\omega|\Theta) \tag{3}$$

The log posterior of this model has a similar lower bound to the variational objective (Eq. 1):

$$\ln p(\Theta|X, Y) \geqslant \sum_{i=1}^{N} \int p(\omega|\Theta) \ln p(y_i|x_i, \omega)d\omega + \ln p(\Theta) - C_{MAP} \tag{4}$$

See Appendix C for detailed derivation. Dropping the normalisation constant $C_{MAP}$ that doesn't depend on $\Theta$, and approximating the above integrals with a single sample, the loss corresponding to the MAP objective becomes:

$$\mathcal{L}_{MAP} = -\sum_{i=1}^{N} \ln p(y_i|x_i, \hat{\omega}_i) - \ln p(\Theta) \tag{5}$$

The first term of this loss is identical to that of the loss for variational dropout (Eq 2). If the prior on $\Theta$ is a zero mean gaussian, then the second term is equivalent to a weight decay penalty just like the KL term in the variational setup. With the two losses being effectively the same, in the following we focus on MAP estimation for the conditional model to sidestep any questions about whether variational inference makes sense in this case.

## 3 THE DROPOUT FAMILY OF MODELS

Having developed a conditional model for dropout that leads to the same objective as variational dropout, we now derive a family of models whose objectives are all lower bounded by the usual dropout objective. We draw inspiration from the different evaluation methods employed for dropout:

- *Deterministic dropout* propagates the expectation of each unit through the network in single pass. This is very efficient and is viewed as a good approximation to the next option.
- *MC dropout* mimics the training procedure, and averages the predicted probabilities over randomly sampled dropout masks. With one forward pass per sample, this can be rather expensive. There is some ambiguity as to what kind of averaging shall be applied: oftentimes the *geometric* average (GMC) is used, because of its close relationship to the loss, but the *arithmetic* average (AMC) is also widespread.

Our goal in this section is to demonstrate the consequences of optimising a lower bound instead of the true objective. While it is easy to argue in general that objectives of more than one model may share any given lower bound, for dropout a particularly simple explicit construction of such a family of models is possible. As we will see, this allows for post-training model selection based on validation results given a trained set of parameters. In the absence of validation results to guide model selection, inspection of the tightness of the lower bound indicates the deterministic model as the most reasonable choice from the family.

### 3.1 GEOMETRIC MODEL

First, we investigate whether the geometric or the arithmetic mean is the correct choice for making predictions in the context of classification. Recall the predictive term of the MAP loss in Eq. 5:

$\sum \ln p(y_i|x_i, \hat{\omega}_i)$. Notice how with SGD and multiple epochs, for each data point several dropout masks are encountered, and the approximating quantity becomes the geometric mean of the predicted probabilities $p(y_i|x_i, \omega)$ over the masked weights. For this reason, the posterior predictive distribution $p(y^*|x^*, X, Y)$ is often computed as the renormalised geometric mean. This is in apparent conflict with the conditional model that prescribes the arithmetic mean (integrating $\omega$ out of Eq. 3). However, we can define another model where the conditional distribution is directly defined to be the renormalised geometric mean

$$p(y|x, \Theta) = \frac{\exp\left(\mathbb{E}_{\hat{\omega} \sim p(\omega|\Theta)} \ln p(y|x, \hat{\omega})\right)}{Z(x, \Theta)}, \quad Z(x, \Theta) = \sum_{c=1}^{C} \exp\left(\mathbb{E}_{\hat{\omega} \sim p(\omega|\Theta)} \ln p(c|x, \hat{\omega})\right) \quad (6)$$

with a slight abuse of notation, due to using the symbol $p$ in $p(y|x, \Theta)$ although $p(y|x, \Theta) \neq \mathbb{E}_{\omega} p(\omega|\Theta) p(y|x, \omega)$. It can be shown that the arithmetic model's (Eq. 3) lower bound (Eq. 4) is a lower bound for this renormalised geometric model (Eq. 6), as well. See Appendix D for the derivation. The answer to the question whether we should use GMC or AMC is that it depends: they correspond to different models, but the dropout objective is a lower bound on the objectives of both models. So one can freely choose between GMC and AMC *at evaluation time*, doing model selection retrospectively after training.

## 3.2 THE POWER MEAN MODEL FAMILY

Having two models to choose from, it is natural to ask whether these are just instantiations of a larger class of models. We propose the power mean family of models to extend the set of models to a continuum between the geometric and arithmetic models described in §3.1 and §2.1, respectively, and show that they have the same lower bound. The power mean is defined as:

$$M_\alpha(x_1, \ldots, x_n) = \left(\frac{1}{n} \sum_{i=1}^{n} x_i^\alpha\right)^{1/\alpha}$$

For $\alpha = 1$ we arrive at the arithmetic mean while the natural extension to $\alpha = 0$ is the geometric mean as it is the limit of $M_\alpha$ at $\alpha \to 0$, which can be proven with L'Hôpital's rule. Similarly to the construction of the geometric model, we define the power mean model by directly conditioning on $\Theta$:

$$p(y|x, \Theta) = \frac{\sqrt[\alpha]{\mathbb{E}_{\hat{\omega} \sim p(\omega|\Theta)} p(y|x, \hat{\omega})^\alpha}}{Z(x, \Theta)}, \qquad Z(x, \Theta) = \sum_{c=1}^{C} \sqrt[\alpha]{\mathbb{E}_{\hat{\omega} \sim p(\omega|\Theta)} p(c|x, \hat{\omega})^\alpha} \quad (7)$$

where $Z(x, \Theta)$ is at most 1 if $\alpha \in (-\infty, 1]$ because $M_\alpha$ is monotonically increasing in $\alpha$ and $Z$ is 1 for $\alpha = 1$. Here we provide a concise derivation of a lower bound on the log posterior (the full derivation can be found in Appendix E):

$$\ln p(\Theta|X, Y) = \sum_{i=1}^{N} \left[ \ln \sqrt[\alpha]{\mathbb{E}_{\hat{\omega} \sim p(\omega|\Theta)} p(y_i|x_i, \hat{\omega})^\alpha} - \ln(Z(x_i, \Theta)) \right] + \ln p(\Theta) - C_{MAP}$$

$$\geqslant \sum_{i=1}^{N} \ln \sqrt[\alpha]{\mathbb{E}_{\hat{\omega} \sim p(\omega|\Theta)} p(y_i|x_i, \hat{\omega})^\alpha} + \ln p(\Theta) - C_{MAP} \quad (8)$$

$$\geqslant \sum_{i=1}^{N} \frac{1}{\alpha} \mathbb{E}_{\hat{\omega} \sim p(\omega|\Theta)} \ln p(y_i|x_i, \hat{\omega})^\alpha + \ln p(\Theta) - C_{MAP} \quad (9)$$

$$= \sum_{i=1}^{N} \int p(\omega|\Theta) \ln p(y_i|x_i, \omega) d\omega + \ln p(\Theta) - C_{MAP}$$

The first inequality above follows from $Z(x, \Theta) \leqslant 1$ for all $x, \Theta$, while the second is an application of Jensen's rule assuming $\alpha > 0$. We arrived at the same lower bound on the objective as we had for the geometric (Eq. 6) and arithmetic models (Eq. 3), thus defining the **power mean family** with parameter $\alpha \in [0, 1]$ of models from which we can choose at evaluation time. For $\alpha > 1$, the normalising constant $Z$ would be greater than 1, and this would not be a lower bound in general.

### 3.3 Tightness of the lower bound

To better understand the quality of fit for models in the power mean family we examine the tightness of their lower bounds. There are two steps involving inequalities in the derivation of the bound: one where the normalisation constant $Z$ is dropped (Eq. 8) and another where the logarithm is moved inside the expectation (Eq. 9). We show that the gaps introduced by these steps can be made arbitrarily small by reducing the variance of $p(y|x, \omega)$ with respect to $\omega$.

Notice that the Jensen gap with the logarithm function is scale invariant:

$$\ln(\mathbb{E}[\lambda L]) - \mathbb{E}\ln(\lambda L) = \ln(\mathbb{E}\, L) - \mathbb{E}\ln(L)$$

Intuitively, this suggests that $\mathrm{var}(L)/(\mathbb{E}\, L)^2$ is closely related to the size of the gap. Indeed, Maddison et al. (2017) show that if the first inverse moment of $L$ is finite, then

$$\ln(\mathbb{E}\, L) - \mathbb{E}\ln(L)) = \frac{\mathrm{var}(L)}{2(\mathbb{E}\, L)^2} + \mathcal{O}(\sqrt{\mathbb{E}[(L - \mathbb{E}\, L)^6]}) \qquad (10)$$

Here we go a bit further and show that if there is a positive lower and upper bound on $L$, then there are non-trivial lower and upper bounds on its Jensen gap and these are bounds are multiplicative in $\mathrm{var}(L)$. Let $L$ be a random variable such that $P(L \in (a, b)) = 1$ where $-\infty \leqslant a < b \leqslant \infty$. Furthermore, let $\varphi(l)$ be a convex function. Jensen's inequality states that $E[\varphi(L)] \geqslant \varphi(\mathbb{E}[L])$. Liao & Berg (2017) show that the Jensen gap $E[\varphi(L)] - \varphi(\mathbb{E}[L])$ can be bounded from below and above:

$$\inf\{h(l; \mu) \mid l \in (a, b)\}\, \mathrm{var}(L) \leqslant \mathbb{E}[\varphi(L)] - \varphi(\mathbb{E}[L]) \leqslant \sup\{h(l; \mu) \mid l \in (a, b)\}\, \mathrm{var}(L)$$

$$h(l; \mu) = \frac{\varphi(l) - \varphi(\mu)}{(l - \mu)^2} - \frac{\varphi'(\mu)}{l - \mu}$$

where $h(l; \mu)$ does not depend on the distribution of $L$, only on its expected value $\mu$ and on the function $\varphi$. Substituting $L = p(c|x, \omega)$ (a random variable on $[0, 1]$ due to the randomness of the dropout masks) and $\varphi(l) = -\ln(l)$, we can see that the gap introduced by Eq. 9 can be made smaller by decreasing the variance of the predictions while maintaining the expected value of $L$ (i.e. the expected probability), assuming that there is a positive lower and upper bound on them (so that the supremum is finite and the infimum is positive, respectively). A similar argument based on $\sum_{c=1}^{C} M_\alpha(\mathbb{E}_{\hat{\omega}}\, p(c|x, \hat{\omega})) = 1$ shows that $Z$ approximately monotonically approaches 1 as the variance decreases, so the gap of Eq. 8 can also be reduced.

Suppose we pick a base model from the power mean family and have a continuum of subvariants with gradually reduced variance in their predictions but the same expectation. Clearly, for each of them we can derive a lower bound the same way as we did for the power mean family. And as we showed above, the lower bounds will *tend to* increase as the variance of the predictions decreases (see Fig. 1a). They do not strictly increase, only tend to, due to how the Jensen gap is bounded from above and below and also due to the $\mathcal{O}$ term of Eq. 10. Nonetheless, as we approach determinism the lower bound is forced into increasingly tighter ranges with strictly monotonically increasing bounds around it, thus we can always reduce the variance such that there is no overlap between the ranges and we get a guaranteed improvement on the lower bound. This effect reaches its apex at the deterministic model whose lower bound is both exact and higher than any other model's. Fig. 1b illustrates that regardless of the choice of base model, reducing the prediction variance will eventually transform it into the same deterministic model.

### 3.4 The extended power mean family: controlling the tightness of the bound

Intuitively, in the absence of other sources of stochasticity the dropout rate controls the variance of the predictions and if it is low, the lower bound can be pretty snug. However, there are two problems.

First, decreasing the dropout rate does not necessarily keep the expectation of the predictions the same. We offer no solution to this bias issue, but refer the reader to previous studies of dropout's approximation properties such as (Baldi & Sadowski 2013) and our subsequent empirical results.

Second, reducing the dropout rate would trade off generalisation for tighter bounds. But doing so only at evaluation time leaves the training time regularisation effect intact, and can be seen as picking another model whose lower bound tends to be higher than that of the base model. Having thus extended the dropout family further, we can now tweak both $\alpha$ and dropout rates at evaluation time.

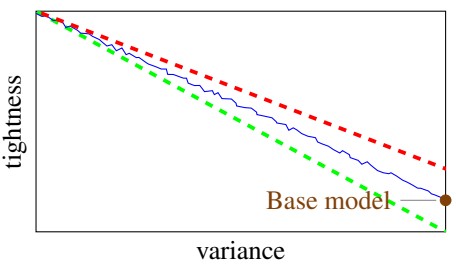 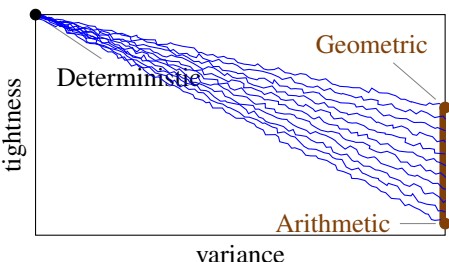

(a) Lower and upper bounds on the lower bound of the model objective as a function of prediction variance for any model in the power mean family.

(b) With reduced variance, all members of the power mean family (brown shading) converge to the deterministic model while their lower bounds tighten.

Figure 1: Tightness of lower bounds vs evaluation time prediction variance in the extended dropout family.

Table 1: PTB training XEs with various dropout rate multipliers between deterministic and GMC. Observe the monotonic improvement in training fit when reducing the dropout rate *at evaluation only*.

| ×0.0 | ×0.1 | ×0.2 | ×0.3 | ×0.4 | ×0.5 | ×0.6 | ×0.7 | ×0.8 | ×0.9 | ×1.0 |
|------|------|------|------|------|------|------|------|------|------|------|
| 2.731 | 2.738 | 2.746 | 2.755 | 2.766 | 2.777 | 2.791 | 2.807 | 2.826 | 2.849 | 2.878 |

Depending on the severity of the introduced bias compared to the benefits of having a tighter lower bound, the optimal variance may lie anywhere between the deterministic and the base model. We show experimentally that across a number of datasets the benefits of tighter bounds matter more, and observe monotonic improvement in model fit as evaluation time dropout rates are decreased all the way to full determinism. The experiment was conducted as follows. On an already trained model, the dropout rate was multiplied by $\lambda \in [0, 1]$. As Table 1 shows, the model fit as measured by cross entropy (XE) on the *training set* improves monotonically when reducing $\lambda$. Results on other datasets and with other power mean models are very similar. We call the union of the reduced dropout rate subvariants of all power mean family models the **extended dropout family** parameterised by $\alpha, \lambda$.

Therefore, we can say that dropout training optimises a deterministic model subject to regularisation constraints, and deterministic evaluation, widely believed to approximate MC evaluation, is the closest match to the true objective at our disposal. It is not that dropout evaluation has a deterministic approximation: dropout trains a deterministic model first and foremost and a continuum of stochastic ones to various extents.

In summary, we described dropout training as optimising a common lower bound for a family of models. Since this lower bound is the same for all models in the family, we can nominate any of them at evaluation time. However, the tightness of the bound varies, which affects model fit. Having trained a model with dropout, the best fit is achieved by the deterministic model with no dropout. This result isolates the regularisation effects from the biases of the lower bound and the dropout family.

## 4 Applying dropout

We investigate how members of the extended dropout model family perform in terms of generalisation. We follow the experimental setup of Melis et al. (2017) and base our work on their best performing model variant for each dataset. Unless explicitly stated, no retraining was performed and their model weights reused. In the experiments with the tuning objective, we follow their experimental setup, using Google Vizier (Golovin et al. 2017), a black-box hyperparameter tuner based on batched Gaussian Process Bandits.

See Table 2 for results of image classification on MNIST, character based language modelling on Enwik8, word based language modelling on PTB and Wikitext-2. On MNIST, deterministic dropout is the best in terms of cross entropy, which matches our theoretical predictions. In contrast, on language modelling arithmetic averaging produces the best results, which necessitates further analysis.

Table 2: Validation XEs on some datasets varying the power $\alpha$ and the dropout rate mulitiplier $\lambda$. Deterministic dropout is not the best evaluation method for the language modelling datasets due to a simple smoothing effect.

| Dataset | DET | Geometric ($\alpha = 0$) | | | Power $\alpha = 0.5$ | | | Arithmetic ($\alpha = 1$) | | |
|---|---|---|---|---|---|---|---|---|---|---|
| | | ×0.8 | ×0.9 | ×1.0 | ×0.8 | ×0.9 | ×1.0 | ×0.8 | ×0.9 | ×1.0 |
| MNIST | **0.070** | 0.087 | 0.087 | 0.088 | 0.92 | 0.93 | 0.93 | 0.100 | 0.100 | 0.100 |
| Enwik8 | 0.886 | 0.879 | 0.878 | 0.881 | 0.877 | 0.877 | 0.877 | 0.875 | 0.875 | **0.875** |
| PTB | 4.110 | 4.090 | 4.090 | 4.093 | 4.072 | 4.070 | 4.073 | **4.061** | 4.064 | 4.080 |
| Wikitext-2 | 4.236 | 4.229 | 4.231 | 4.235 | 4.025 | 4.026 | 4.208 | **4.203** | 4.212 | 4.228 |

Table 3: PTB training and validation XEs for AMC at $\lambda \in \{0, 0.8, 1\}$ per word frequency. Note how DET dominates AMC on the training set, but AMC is better for rare words in the validation set.

| frequency | num targets | training DET | ×0.8 | AMC | validation DET | ×0.8 | AMC |
|---|---|---|---|---|---|---|---|
| 25000< | 13580 | 1.40 | 1.50 | 1.56 | 1.58 | 1.64 | 1.68 |
| 5000< | 26658 | 1.65 | 1.75 | 1.81 | 1.93 | 1.98 | 2.02 |
| 500< | 44702 | 2.19 | 2.30 | 2.36 | 2.58 | 2.63 | 2.66 |
| <500 | 29058 | 4.07 | 4.19 | 4.29 | 6.49 | 6.39 | 6.39 |
| <100 | 14222 | 4.24 | 4.38 | 4.49 | 7.81 | 7.64 | 7.61 |
| <20 | 5008 | 4.00 | 4.19 | 4.33 | 9.20 | 9.01 | 8.97 |

We suspected that the particularly severe form of class imbalance exhibited by the power-law word distribution (Zipf 1935) might play a role. To verify this, we contrasted training and validation XEs on PTB for words grouped by frequency (see Table 3). On the training set, the gap between deterministic dropout and AMC is wider for low frequency words. On the validation set, AMC is worse for frequent words but better for rare words. The ×0.8 dropout multiplier just finds a reasonable compromise.

## 4.1 SOFTMAX TEMPERATURE

The observed effect is consistent with smoothing, thus we posit that the reason MNIST results are worse with AMC is that the marginal distributions of labels in the training and test set are identical by construction and further smoothing is unnecessary. On the other hand, PTB and Wikitext-2 benefit from AMC's smoothing because the penalty for underestimating low probabilities is harsh, hence the large improvement on rare words. The character based Enwik8 dataset lies somewhere in between: the training and test distributions are better matched and there are no very low probability characters.

To test the hypothesis that AMC's advantage lies in smoothing, we tested how performing smoothing by other means affects the results. In this experiment, on a trained model the temperature of the final

Table 4: Validation and test perplexities on PTB and Wikitext-2 with various evaluation strategies and default or optimal validation softmax temperatures. Our baseline results correspond to DET at temperature 1. Note that AMC does not benefit from setting the optimal softmax temperature ("opt"), while DET is improved by it almost to the point of matching AMC which supports the smoothing hypothesis.

| | Dataset | Temp | DET | Geometric ($\alpha = 0$) | | | Power $\alpha = 0.5$ | | | Arithmetic ($\alpha = 1$) | | |
|---|---|---|---|---|---|---|---|---|---|---|---|---|
| | | | | ×0.8 | ×0.9 | ×1.0 | ×0.8 | ×0.9 | ×1.0 | ×0.8 | ×0.9 | ×1.0 |
| Validation | WT-2 | 1 | 69.1 | 68.6 | 68.8 | 69.1 | 67.0 | 67.2 | 67.2 | **66.9** | 67.5 | 68.6 |
| | | opt | **67.4** | **67.5** | 67.7 | 68.0 | **67.0** | 67.1 | 67.2 | 66.9 | 67.4 | 68.1 |
| | PTB | 1 | 60.9 | 59.6 | 59.7 | 59.7 | 58.1 | 57.9 | 58.0 | 57.3 | 57.5 | 58.5 |
| | | opt | **57.5** | **57.5** | 57.9 | 58.3 | **57.1** | 57.3 | 57.8 | **57.1** | 57.5 | 58.4 |
| Test | WT-2 | 1 | 65.9 | 65.3 | 65.4 | 65.6 | **63.8** | 63.9 | 64.2 | **63.7** | 64.5 | 65.5 |
| | | opt | **64.5** | **64.7** | 64.8 | 64.9 | 63.8 | 63.8 | 64.2 | 63.7 | 64.2 | 64.9 |
| | PTB | 1 | 58.6 | 57.3 | 57.4 | 57.4 | 56.0 | 55.8 | 55.9 | **55.3** | 55.5 | 56.5 |
| | | opt | **56.0** | **56.0** | 56.1 | 56.5 | 55.7 | **55.7** | 56.0 | 55.3 | 55.5 | 56.3 |

softmax was optimised on the validation set and the model was applied with the optimal temperature to the validation and test sets. Our experimental results in Table 4 support the hypotheses that AMC smooths the predicted distribution as increasing the temperature improves DET and GMC considerably but not AMC. In fact, the optimal temperature for AMC with $\lambda = 1$ was slightly lower than 1, which corresponds to sharpening, not smoothing.

Tuning the evaluation time softmax temperature is similar to label smoothing (Pereyra et al. 2017), the main difference being that our method does not affect training. While this is convenient, for tuning model hyperparameters, ideally we would determine the optimal evaluation parameters $\alpha$, $\lambda$ and the temperature for the calculation of the validation score for each set of hyperparameters tried, but this would be prohibitively expensive. Since deterministic evaluation coupled with the optimal temperature is very close to the best performing AMC model, it serves as a good proxy for the ideal tuning objective. The optimal temperature can be approximately determined using a linear search on a subset of the validation data which is orders of magnitude faster than MC dropout. In our experiments, hyperparameter tuning with validation scores computed at the optimal softmax temperature did improve results, albeit very slightly (about half a perplexity point). Thus we can conclude that deterministic dropout is already a reasonable proxy for which to optimise.

## 4.2 RESULTS

We have improved the best test result of Melis et al. (2017) from **58.3** to **55.7** on PTB, and from **65.9** to **63.7** on Wikitext-2 using their model weights, only tuning the evaluation parameters $\alpha$, $\lambda$ and the softmax temperature on the validation set. By retuning the hyperparameters of the PTB model with optimal temperature deterministic evaluation, we improved to **55.3** on PTB. For lack of resources, we did not retune for Wikitext-2. For comparison, the state of the art in language modelling without resorting to dynamic evaluation or a continuous cache pointer is Mixture of Softmaxes (Yang et al. 2017) with 54.44 and 61.45 on PTB and Wikitext-2, respectively. At present, it is unclear whether the benefits of their approach and ours combine.

In summary, we looked at how different models and evaluation methods rank in terms of generalisation. Across a number of tasks and datasets the ranking differed from what was observed on the training set. We found that AMC smooths the distribution of the prediction probabilities and we achieved a similar effect without resorting to expensive sampling simply by adjusting the temperature of the final softmax. Finally, we brought the tuning objective more in line with the improved evaluation by automatically determining the optimal softmax temperature when evaluating on the validation set which further improved results.

## 5 IMPLICATIONS

The construction of a conditional model family with a common lower bound on their objectives is applicable to other latent variable models with similar structure and inference method. This lower bound admits ambiguity as to what model is being fit to the data, which in turn allows for picking any such model at evaluation time. However, the tightness of the bound and the quality of the fit varies. For dropout, the deterministic model has the best fit even though the training objective is highly stochastic, but this result hinges on the approximation properties of deterministic dropout and will not carry over to other probabilistic models in general. In particular, standard VAEs (Kingma & Welling 2013) with their lower bound being very similar in construction to Eq. 1 cannot quite collapse to a deterministic model else they suffer an infinite KL penalty. Still, the lower bound being looser on the tails of $q$ is related to problem of underestimating posterior uncertainty (Turner & Sahani 2011).

In related works, expectation-linear dropout (Ma et al. 2016) and fraternal dropout (Zolna et al. 2017) both try to reduce the "inference gap": the mismatch between the training objective and deterministic evaluation. The gains reported in those works might be explained by reducing the bias of deterministic evaluation and also by encouraging small variance in the predictions and thus getting tighter bounds. Another recent work, activation regularisation (Merity et al. 2017), could be thought of as a mechanism to reduce the variance of predictions to a similar effect. In the context of language modelling, the connection between noise and smoothing was established by Xie et al. (2017). Our improved understanding further emphasises that connection, and at the same time challenges the way we think about dropout.

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

## APPENDIX A  VARIATIONAL DROPOUT WITH NON-SHARED MASKS

If $q$ and $p$ are redefined for the non-shared setting to be products of identical and independent, per time step factors, neither term of the variational objective requires rethinking: the MC approximation still works since $q$ is easy to sample from, while the KL term becomes a sum of componentwise KL divergences and can still be implemented as weight decay. Consequently, both shared and non-shared masks fit into the variational framework. For a detailed derivation see Appendix B.

In related works, Pachitariu & Sahani (2013) in their investigation of regularisation of standard RNN based language models dismiss applying dropout to recurrent connections "*to avoid introducing instabilities into the recurrent part of the LMs*". Bayer et al. (2013) echo this claim about RNNs, which is then cited by Zaremba et al. (2014), but their work is based on LSTMs not standard RNNs. Finally, Gal & Ghahramani (2016b) cite all of the above but also work with LSTMs. Their results indicate a large, about 15 perplexity point advantage to shared mask dropout for language modelling on the Penn Treebank (PTB) corpus (see Fig. 2 in their paper).

Our experimental results obtained with careful and extensive hyperparameter tuning, listed in Table 5, indicate only a small difference between the two which is in agreement with the empirical study of Semeniuta et al. (2016).

Table 5: Validation and test set perplexities on PTB with shared (S) or non-shared (NS) dropout masks for a small, 1 layer and a large, 4 layer LSTM with 10 and 24 million weights, respectively. Non-shared masks perform nearly as well as shared masks and as we have seen neither is "more variational" than the other.

| dataset | 10M | | 24M | |
|---|---|---|---|---|
| | S | NS | S | NS |
| validation | 59.4 | 60.2 | 57.5 | 58.3 |
| test | 57.5 | 58.6 | 56.0 | 56.9 |

In any case, non-shared masks, in addition to being variational, are also surprisingly competitive with shared masks for LSTMs (we make no claims about standard RNNs). We also tested whether embedding dropout (in which dropout is applied to entire vectors in the input embedding lookup table) proposed by Gal & Ghahramani (2016b) improves results, and find that embedding dropout does not offer any improvement on top of input dropout.

## APPENDIX B  DERIVATION OF VARIATIONAL DROPOUT WITH NON-SHARED MASKS

In this section, we formulate naive (i.e. non-shared mask) dropout in the variational setting. In contrast to the shared mask case, where $\omega$ was a single set of weights, here $\omega^{1:T}$ (or $\omega$, for short) has a set of weights for each time step that differ in their dropout masks. The variational posterior $q(\omega)$ and the prior $p(\omega)$ are both products of identical distributions over time:

$$q(\omega^{1:T}) = \prod_{t=1}^{T} q'(\omega^t) = \prod_{t=1}^{T} \left[ p\mathcal{N}(\omega^t|0, \sigma^2) + (1-p)\mathcal{N}(\omega^t|\Theta, \sigma^2) \right]$$

$$p(\omega^{1:T}) = \prod_{t=1}^{T} p'(\omega^t) = \prod_{t=1}^{T} \mathcal{N}(\omega^t|0, \sigma_p^2)$$

An unbiased approximation to the integrals in Eq. 1 is based on a single, easy to obtain sample $\hat{\omega} \sim q(\omega)$:

$$\int q(\omega) \ln p(y|x, \omega) d\omega \approx \ln p(y|x, \hat{\omega})$$

Showing that the KL term can still be approximated with weight decay with non-shared masks is not much more involved. Both distributions are products of densities over independent random variables,

so the componentwise KL divergencies sum. In particular:

$$
\begin{aligned}
\mathrm{KL}(q(\omega)\|p(\omega)) &= \int \left( \prod_{i=1}^{T} q'(\omega^i) \right) \ln \frac{\prod_{t=1}^{T} q'(\omega^t)}{\prod_{t=1}^{T} p'(\omega^t)} d\omega \\
&= \int \left( \prod_{i=1}^{T} q'(\omega^i) \right) \sum_{t=1}^{T} \ln \frac{q'(\omega^t)}{p'(\omega^t)} d\omega \\
&= \sum_{t=1}^{T} \int \left( \prod_{i=1}^{T} q'(\omega^i) \right) \ln \frac{q'(\omega^t)}{p'(\omega^t)} d\omega \\
&= \sum_{t=1}^{T} \int \left[ \prod_{i=1,i\neq t}^{T} q'(\omega^i) \right] \left[ q'(\omega^t) \ln \frac{q'(\omega^t)}{p'(\omega^t)} \right] d\omega \\
&= \sum_{t=1}^{T} \left[ \int \prod_{i=1,i\neq t}^{T} q'(\omega^i) d\omega^{\backslash t} \right] \left[ \int q'(\omega) \ln \frac{q'(\omega)}{p'(\omega)} d\omega \right] \\
&= T \cdot \mathrm{KL}(q'(\omega)\|p'(\omega))
\end{aligned}
$$

We partitioned the variables into two mutually exclusive sets $w^t$ and its complement $w^{\backslash t}$, and split the multiple integral using Fubini's theorem (or, equivalently, using the expectation of independent random variables rule). After the split, the first integral is trivially 1 and the second has no dependence on $T$.

What we end up with is a sum of identical KL terms of the same distributions as in the shared mask case, so the full KL can be approximated with weight decay.

## APPENDIX C   DERIVATION OF THE MAP LOWER BOUND FOR THE ARITHMETIC MODEL

We can rewrite the posterior as:

$$
\begin{aligned}
p(\Theta|X,Y) &= \frac{p(X,Y|\Theta)p(\Theta)}{p(X,Y)} \\
&\propto p(X,Y|\Theta)p(\Theta) \\
&= \int p(Y|X,\omega,\Theta)p(\omega|\Theta,X)p(\Theta|X)p(X)d\omega \\
&\propto \int p(Y|X,\omega)p(\omega|\Theta)p(\Theta)d\omega
\end{aligned}
$$

Moving to the log domain and using Jensen's inequality allows us to construct a lower bound that is a sum of per data point terms (i.e. something that can be conveniently optimised):

$$
\begin{aligned}
\ln p(\Theta|X,Y) &= \ln \int p(Y|X,\omega)p(\omega|\Theta)p(\Theta)d\omega - C_{MAP} \\
&= \ln \int p(\omega|\Theta) \prod_{i=1}^{N} p(y_i|x_i,\omega)d\omega + \ln p(\Theta) - C_{MAP} \\
&\geqslant \int p(\omega|\Theta) \ln \prod_{i=1}^{N} p(y_i|x_i,\omega)d\omega + \ln p(\Theta) - C_{MAP} \\
&= \sum_{i=1}^{N} \int p(\omega|\Theta) \ln p(y_i|x_i,\omega)d\omega + \ln p(\Theta) - C_{MAP}
\end{aligned}
$$

## APPENDIX D    DERIVATION OF THE MAP LOWER BOUND FOR THE GEOMETRIC MODEL

From Eq. 6 recall that:

$$p(y|x, \Theta) = \frac{\exp\left(\mathbb{E}_{\hat{\omega} \sim p(\omega|\Theta)} \ln p(y|x, \hat{\omega})\right)}{Z(x, \Theta)}$$

The normalisation constant $Z$ is at most 1, due to the geometric mean being bounded from above by the arithmetic mean on a per class $c$ basis:

$$Z(x, \Theta) = \sum_{c=1}^{C} exp\left(\mathbb{E}_{\hat{\omega} \sim p(\omega|\Theta)} \ln p(c|x, \hat{\omega})\right)$$

$$\leqslant \sum_{c=1}^{C} \mathbb{E}_{\hat{\omega} \sim p(\omega|\Theta)} p(c|x, \hat{\omega})$$

$$= \mathbb{E}_{\hat{\omega} \sim p(\omega|\Theta)} \sum_{c=1}^{C} p(c|x, \hat{\omega}) = 1$$

Since this a conditional model, we can rewrite the posterior as:

$$p(\Theta|X, Y) = \frac{p(X, Y|\Theta)p(\Theta)}{p(X, Y)}$$

$$\propto p(X, Y|\Theta)p(\Theta)$$

$$= p(Y|X, \Theta)p(X|\Theta)p(\Theta)$$

$$\propto p(Y|X, \Theta)p(\Theta)$$

$p(X|\Theta)$ is dropped in the last step as it is constant. Moving to the log domain once again:

$$\ln p(\Theta|X, Y) = \ln p(Y|X, \Theta) + \ln p(\Theta) - C_{MAP}$$

$$= \ln \prod_{i=1}^{N} p(y_i|x_i, \Theta) + \ln p(\Theta) - C_{MAP}$$

$$= \sum_{i=1}^{N} \left[ \mathbb{E}_{\hat{\omega} \sim p(\omega|\Theta)} \ln p(y_i|x_i, \hat{\omega}) - \ln(Z(x_i, \Theta)) \right] + \ln p(\Theta) - C_{MAP}$$

$$\geqslant \sum_{i=1}^{N} \mathbb{E}_{\hat{\omega} \sim p(\omega|\Theta)} \ln p(y_i|x_i, \hat{\omega}) + \ln p(\Theta) - C_{MAP}$$

$$= \sum_{i=1}^{N} \int p(\omega|\Theta) \ln p(y_i|x_i, \omega) d\omega + \ln p(\Theta) - C_{MAP}$$

where the lower bound arises due to $\forall i \colon Z(x_i, \Theta) \leqslant 1$.

## APPENDIX E    DERIVATION OF THE MAP LOWER BOUND FOR THE POWER MEAN FAMILY

In §3.2 we proved that $\forall i \colon Z(x_i, \Theta) \leqslant 1$. Starting from $p(\Theta|X, Y) \propto p(Y|X, \Theta)p(\Theta)$ just like in the geometric case, we derive a lower bound in the log domain:

$$\ln p(\Theta|X, Y) = \ln p(Y|X, \Theta) + \ln p(\Theta) - C_{MAP}$$

$$= \ln \prod_{i=1}^{N} p(y_i|x_i, \Theta) + \ln p(\Theta) - C_{MAP}$$

$$= \sum_{i=1}^{N} \left[ \ln \sqrt[\alpha]{\mathbb{E}_{\hat{\omega} \sim p(\omega|\Theta)} p(y_i|x_i, \hat{\omega})^{\alpha}} - \ln(Z(x_i, \Theta)) \right] + \ln p(\Theta) - C_{MAP}$$

$$\geqslant \sum_{i=1}^{N} \ln \sqrt[\alpha]{\mathop{\mathbb{E}}_{\hat{\omega} \sim p(\omega|\Theta)} p(y_i|x_i, \hat{\omega})^{\alpha}} + \ln p(\Theta) - C_{MAP}$$

$$= \sum_{i=1}^{N} \frac{1}{\alpha} \ln \left( \mathop{\mathbb{E}}_{\hat{\omega} \sim p(\omega|\Theta)} p(y_i|x_i, \hat{\omega})^{\alpha} \right) + \ln p(\Theta) - C_{MAP}$$

$$\geqslant \sum_{i=1}^{N} \frac{1}{\alpha} \mathop{\mathbb{E}}_{\hat{\omega} \sim p(\omega|\Theta)} \ln p(y_i|x_i, \hat{\omega})^{\alpha} + \ln p(\Theta) - C_{MAP}$$

$$= \sum_{i=1}^{N} \mathop{\mathbb{E}}_{\hat{\omega} \sim p(\omega|\Theta)} \ln p(y_i|x_i, \hat{\omega}) + \ln p(\Theta) - C_{MAP}$$

$$= \sum_{i=1}^{N} \int p(\omega|\Theta) \ln p(y_i|x_i, \omega) d\omega + \ln p(\Theta) - C_{MAP}$$

