# OpenReview forum: "Pushing the bounds of dropout"
_ICLR.cc/2019/Conference_

### Official Review · AnonReviewer1 · 2018-11-02
**Try to provide new perspectives on the problem. But they are not sound. No convincing approach.**

**Rating:** 4
**Confidence:** 3

**Review:**

This paper studies the problem of making predictions with a model trained using dropout. Authors try to provide a theoretical foundation for using dropout when making predictions. For this purpose, they show that when using dropout training we are maximizing a common lower bound on the objectives of a family of models, including most of the previously used methods for prediction with dropout.

I find that the paper addresses a relevant problem and try to apply a novel approach. But, in general, I find the paper is not easy to follow and to grasp the main ideas.

Here I detail my main concerns:


1. This is one of my main concerns. The contraposition between the geometric and the average model. I don't like this contraposition. The average model is just the standard marginalization operation over the weights, $p(y|x) = \int p(y|x,w)p(w|\Theta)dw$. This is the natural solution for the prediction problem to the problem if we accept the generative model given in Eq (3).

In the case of the variational dropout, we depart from the same generative model, but we employ an approximation. It is the variational approximation the one that induces the geometric mean provided in eq (6). I.e. if we want to compute the posterior over the label y* for a sample x*, after training, we should compute the associated lower bound
$\ln p(y*|x*) >= E_q[\ln p(y*|x*,w)] - KL(q|p)$
In this case, q(w) = p(w|\Theta), as stated Eq (3) and in the corresponding equation provided in page 2 (the q(w) is not learnt because it only depends on the dropout rate, while the $\Theta$ are learnt by maximum log-likelihood and do not have a $q$ associated).  This gives rise to the geometric mean approximation provided in Eq (6).  I.e. the geometric mean prediction is simply the result of using a variational approximation at prediction time.

My problem here is that authors employ convoluted arguments to introduce this geometric mean prediction and the average prediction, without making the connection discussed above.

3. Section 3.3 and 3.4 introduces new arguments for modifying the dropout rate (and the alpha) parameter at test time. But, again, I find the arguments convoluted. We consider the dropout rate a hyper-parameter of the model, the standard learning theory tells us to fix the parameters with the training data and evaluate them later when making predictions. Why should we use different dropout rates at training and testing? Authors arguments about the tightness of the bound of Eq (8) and Eq(9). are not convincing to me.

So, I don't find authors provide convincing answers to the raised questions at the beginning of the paper about the use of dropout when making predictions.

Minor comments:

1. The generative model for Variational dropout is the same than the generative model for the "conditional model", eq. (3).

2. In Eq. (7) authors are defining the weighted power mean. I think it would be clearer to directly introduce the weighted power mean instead of the standard power mean in Section 3.2.

3. Section 3.3. I find some parts are difficult to understand. "suppose we pick a base model from the power mean family and have a continuum of subvariants with gradually reduced variance in their predictions but the same expectation." Later, I can understand authors are referring to the possibility of reducing the dropout rate.

---

> ### Author Response · Authors · 2018-11-12
> **Re: Try to provide new perspectives on the problem. But they are not sound. No convincing approach.**
>
> We thank reviewer #1 for their comments.
>
> 1. We agree that marginalizing out the weights is the natural thing to do if we accept the generative model. However, due to the training objective being a lower bound on the true objectives of the proposed family of models why would we prefer the arithmetic model to the others? This one of the main points of the paper.
>
>  As to the variational approximation leading to the geometric mean prediction, they are closely related but the variational approximation lacks the normalizing constant Z present in eq 6. We acknowledge this relationship by saying that "oftentimes the geometric average (GMC) is used, because of its close relationship to the loss".
>
> 3. There is no disagreement with standard learning theory, in a sense we propose a shortcut: there is no need to train the model for different alpha, lambda choices since these hyperparameters only affect evaluation. This is due to the training objective being a lower bound on the true objectives of the proposed family of models.
>
> It is unclear what is unconvincing about the tightness of the bound argument. In any case, our intention there is to argue that in the absence of validation data, the most natural evaluation method is the deterministic one. However, if there is validation data available, then the tightness of the bound argument is not necessary: tuning alpha, lambda only requires the common lower bound.
>
>
> In general, regarding the paper not being convincing, sound and the arguments being convoluted, we would like to improve both content and presentation. To that end, we ask for more detailed feedback on these points, if possible.

---

### Official Review · AnonReviewer3 · 2018-11-04
**Interesting paper but still lack of novelty**

**Rating:** 5
**Confidence:** 2

**Review:**

The paper point that the dropout in training is equivalent to MAP estimate of hierarchical models when the prior distribution of weights, \Theta, is a zero mean Gaussian. Based on that observation the authors propose several different evaluation methods for dropout. The experimental results show that the proposed evaluation methods improved the performance of language models.

Are there any experimental results of the proposed evaluation methods for another type of data beyond language modeling?

Do the term "deterministic dropout" in the last sentence of the first paragraph on page 1 and the one in Sec 3 (the first bullet) refer to the same thing?

Minor: gaussian -> Gaussian

---

> ### Author Response · Authors · 2018-11-12
> **Re: Interesting paper but still lack of novelty**
>
> We feel that the paper has plenty of novelty. Here is a list of our contributions:
> - theoretical: dropout training optimizes a common lower bound for a family of models
> - theoretical: in the absence of validation data, we should choose the deterministic model, because it has the highest and tightest lower bound
> - theoretical: evaluation-time model selection (or lazy hyperparameter tuning)
> - empirical: demonstrating that one important side effect of MC dropout is smoothing (i.e. tweaking the softmax temperature)
> - empirical: improved language modelling results (even using weights of heavily tuned models)
>
> Currently, we only have language modelling results. Preliminary experiments on MNIST indicated no benefit to our method. In section 4, we provide an analysis and argue that class imbalance is a crucial factor.
>
> Yes, deterministic dropout refers to the same thing at both places.

---

### Official Review · AnonReviewer4 · 2018-11-10
**A special interpretation of Dropout. Unfortunately, not convincing.**

**Rating:** 5
**Confidence:** 3

**Review:**

Different from an existing variational dropout method which used variational inference to explain Dropout, this paper proposes to interpret Dropout from the MAP perspective. More specifically, the authors utilize the Jensen inequality to develop a lower bound for log-posterior, which is used as training objective for dropout. They then exploit the power mean to develop the conditional power mean model family, which provide additional flexibility for evaluation during validation.
Even though the way how the proposed method is analyzed/generalized is interesting, the proposed method is not convincing, but I am not absolutely sure. Besides the paper is hard to follow, some other concerns are listed below.
(1) “…the original/usual dropout objective” and “the dropout rate” are not defined in the paper, even though they appear many times in the paper.
(2) In the last paragraph of Sec. 2, the authors argue that utilizing their MAP objective “sidestep any questions about whether variational inference makes sense.” However, the presented MAP lower bound has its own problem, since it is derived using the Jensen inequality.  For example, as shown in Appendix C, the equality becomes true only when p(w|\Theta) is a delta function.
(3) How to tune the hyperparameters (alpha, lambda) of the extended dropout family in practice?
(4) The current experiments might be weak. Additional experiments on popular image datasets are recommended.

Minors:
(1) In Eq. (3), is p(w_r|\Theta) of the second formula identical to p(w|\Theta) of the third formula?
(2) In the second row below Eq. (6), E_w p(w|\Theta) p(y|x,w) is a typo.

---

> ### Author Response · Authors · 2018-11-12
> **Re: A special interpretation of Dropout. Unfortunately, not convincing.**
>
> Main criticism is that the method not convincing. Does that refer to the theoretical or empirical results or both? What are the areas we should improve?
>
> (1) We'll address this oversight in an upcoming revision.
>
> (2) It is true that all objectives derived in the paper are lower bounds and so is the variational bound. They are also the same. Our goal with the MAP construction is not to have an exact objective but to avoid the question whether it makes sense to use a variational approach purely for regularizing a model (see the beginning of section 2.1 and Osband, 2018).
>
> (3) Alpha and lambda were tuned with a simple grid search (not unlike table 4). We will clarify this in the paper.
>
> (4) In our experiments, imbalance of class label distribution is shown to be the crucial for the Monte-Carlo evaluation method to perform better than deterministic evaluation (see table 2, especially the MNIST results). Thus there is no reason to expect that large image datasets - on which dropout regularization is not a big win to start with - would benefit from our method.

---

> > ### Comment · AnonReviewer4 · 2018-11-26
> > **Comments**
> >
> > Thanks for your response. However, I don't agree with your response in (2), which is also highly related to my main concern of the method being not convincing.
> >
> > In fact, your MAP lower bound and the variational lower bound are not the same.
> > (i) The variational lower bound is tight when the variational posterior approaches the true posterior;
> > (ii) by contrast,  your MAP lower bound is tight only when the prior is a delta function, that is when variable w becomes deterministic. That also explains why the deterministic model has the tightest lower bound.
> >
> > The aforementioned point is not considered well addressed currently, accordingly comes my main concern.

---

> > > ### Author Response · Authors · 2018-12-05
> > > **Re: Comments**
> > >
> > > Thank you for the followup. What we intend to show in the paper is that the actual loss being optimized for models with dropout is an approximation in the same vein to the variational and MAP objectives (for all members of family). If the MAP formulation explains why the deterministic variant is best, then the same argument applies to the variational model when optimized in that particular way.

---

### Meta-Review · Area_Chair1 · 2018-12-11
**New perspectives on dropout but unconvincing arguments**

**Confidence:** 5
**Recommendation:** Reject

**Metareview:**

The paper tried to introduce a new interpretation of dropout and come with improved algorithms. However, the reviewers were not convinced that the presented arguments were correct/novel, and they found the paper difficult to follow. The authors are encouraged to carefully revise their paper to address these concerns.